# Evaluation of Correlation between Sleep and Psychiatric Disorders in a Population of Night Shift Workers: A Pilot Study

**DOI:** 10.3390/ijerph20043756

**Published:** 2023-02-20

**Authors:** Luigi Cirrincione, Fulvio Plescia, Ginevra Malta, Marcello Campagna, Luigi Isaia Lecca, Alenka Skerjanc, Elisa Carena, Vincenzo Baylon, Kelly Theodoridou, Santo Fruscione, Emanuele Cannizzaro

**Affiliations:** 1Department of Health Promotion Sciences Maternal and Child Care, Internal Medicine and Medical Specialties ‘Giuseppe D’Alessandro’, University of Palermo, Via del Vespro 133, 90127 Palermo, Italy; 2Department of Medical Sciences and Public Health, University of Cagliari, 09127 Cagliari, Italy; 3Clinical Institute for Occupational, Traffic and Sports Medicine, University Medical Centre Ljubljana, 1000 Ljubljana, Slovenia; 4Department of Sciences of Public Health and Pediatrics, University of Turin, 10126 Turin, Italy; 5Newton Lewis Institute Scientific Research-Life Science Park, 3000 San Gwann, Malta; 6Department of Microbiology, Andreas Syggros University Hospital Athens Greece, 10552 Athens, Greece

**Keywords:** work, insomnia, psychiatric disorders, anxiety, depression shift work, night work, GABA, stress, prevention

## Abstract

Background: Insomnia is the perception of inadequate, insufficient or non-restorative sleep. Of all sleep-related disorders, insomnia is the most common. It is important to remember that the sleep–wake cycle also plays a central role in the genesis of anxiety and depression. The aim of our study is to evaluate the association between sleep disturbances and anxiety and depression in a group of workers of both sexes who perform night shift work. Methods: Information on sleep disorders was collected by administering the Insomnia Severity Index (ISI) questionnaire. Statistical analysis was conducted using the Chi-square test to assess whether there were any differences between sex for those who were healthy or who were diagnosed with psychiatric disorders. Results: The results showed that there was a good percentage of subjects with insomnia problems, impairing normal daily activities and promoting the onset of fatigue, daytime sleepiness, cognitive performance deficits and mood disorders. Conclusion: We highlighted how anxious and depressive anxiety disorders are more pronounced in people who suffer from altered sleep–wake rhythms. Further research in this direction could prove to be fundamental for understanding the genesis of the onset of other disorders as well.

## 1. Introduction

Insomnia is the perception of inadequate, insufficient or non-restorative sleep. Of all sleep-related disorders, insomnia is the most common, with estimates ranging from 30 to 35 per cent; and the prevalence of chronic or severe insomnia is estimated at 10 to 15 per cent [1,2].

Gender, age and life style affect the onset of sleep disorders: it has been observed, for example, that nicotine-dependent individuals have a greater chance of developing this disorder [3] and that the prevalence has also been found to be correlated with the use of certain substances of abuse including alcohol [4,5].

In particular, it has been observed that women have a higher rate of insomnia than men and that these disorders increase with age [6].

It has been observed that there is a correlation between insomnia and hypersomnia and certain systemic diseases. Sleep-disordered individuals have a significantly higher risk of developing cardiovascular diseases such as hypertension, cardiac and cerebral ischemic episodes and psychiatric diseases, among others [7]. The latter include stress, anxiety, depression and, albeit with different pathophysiological mechanisms, post-traumatic stress (PTDS) [8].

Insomnia is a very common condition among people suffering from depression [9]. While approximately 15 per cent of depressed persons sleep more than necessary, 80 per cent report difficulty falling asleep or maintaining sleep [10]. However, the relationship between sleep and depressive disorder is still considered complex.

Depression can promote the genesis of sleep–wake rhythm disturbances that, in turn, may contribute to the exacerbation of the disorder [11]. For some people, depressive symptoms occur before the onset of a sleep disorder; for others, the opposite occurs [10].

The presence of insomnia in depression is, in most cases, associated with a more severe depressive picture. Different research has shown that people with insomnia have a three times greater risk of developing a major depressive episode than those with good sleep quality.

Certain work activities also seem to influence the onset of pathologies related to the alteration of the sleep–wake cycle [12,13,14,15,16,17], both in relation to factors, such as the organisation and the environment in which the work is performed, and to specific conditions that contribute to the increase in stress conditions. In particular, work that includes night shifts (all those jobs that involve working for at least three hours in the interval between midnight and 5 a.m.) seems to be able to negatively influence the individual’s ability to functionally adapt to the circadian rhythm, playing a fundamental role in the genesis of different pathologies [18,19,20].

Although the importance of stress as a pathogenetic factor of various psychosomatic diseases has been recognised since the first half of the last century, it is only in the last few decades that the deepening of knowledge in the fields of neurochemistry, neuroendocrinology and immunology have made it possible to understand and interpret, to a large extent, the variations in the specific mechanisms underlying pathologies induced by an excess of adverse stimuli [21,22].

Adaptive responses to stress can be quantitatively and qualitatively different, depending on the personality and experiences of the individual, his or her biorhythms and the intrinsic characteristics of stressors, such as regularity, predictability, avoidance, duration and intensity, as well as different environmental factors.

On the basis of the above premises, the aim of our pilot study is to assess the association between sleep disorders and psychiatric disorders, specifically anxiety and depression, in a group of workers of both sexes performing different work activities involving night shifts for at least 50 nights per year in order to evaluate possible effective interventions in preventing these disorders.

## 2. Materials and Methods

### 2.1. Study Population

A sample of workers working night shifts for at least 50 nights per year was selected from a private practice of Occupational Medicine in Palermo. This cross-sectional study was conducted from January to September 2022. The selected population initially consisted of 523 (100%) persons of both sexes working in different occupations (employees in healthcare facilities, production workers, parcel sorters, commercial vehicle drivers and passenger transport workers).

Authorisation to participate in this study was given by the employer and/or the company human resources department. Of these, 47 (8.99%) declined to participate; consequently, the study initially recruited 476 workers (91.01%). All participants were informed about the purpose of the research and signed the informed consent before participating. Respondents were asked not to mention their name or the name of their organisation in the questionnaire to ensure privacy and anonymity.

At the first visit, an in-depth anamnesis was conducted on all the workers, including an interview with a neurology specialist, which allowed us to make the differential diagnosis of sleep disorders and to confirm the possible clinical diagnosis of anxious and/or anxious depressive state through the analysis of the documentation submitted.

In addition, the entire sample was subjected to a thorough physical examination, including the assessment of anthropometric (height, weight, body mass index) and cardiovascular (blood pressure and electrocardiogram) parameters.

At the end of the medical evaluation, 27 subjects were excluded from the study because at the time of the research they were taking drugs that could affect sleep quality, different from those used for the possible treatment of anxiety and depression. Finally, a further 36 subjects were not included due to different causes: body mass index greater than 32, dysmetabolic diseases, treatment with antineoplastic drugs, immunosuppressants or corticosteroids.

In the end, our sample consisted of 413 adults, 222 males and 191 females (M/F ratio 1.2).

All data were managed according to the Italian law for the protection of privacy (Decree no. 196, January 2003). A multidisciplinary team of health experts collected and analysed the data obtained through the questionnaire administered on sleep disturbance and also verified the presence or absence of anxiety disorders or depression through the patients’ careful medical history.

### 2.2. Assessment of Sleep Disorders

The collection of information on sleep disturbances was carried out through the administration of the Insomnia Severity Index (ISI) questionnaire, a valuable tool for understanding the nature, severity and impact of insomnia [23]. The time period covered by the test is the ‘last month’, and the parameters assessed are: severity of sleep onset problems, sleep maintenance and morning awakening, sleep dissatisfaction, interference of sleep difficulties with normal daily activities, perception of sleep problems by others and distress caused by poor and unsatisfactory sleep. This instrument consists of a survey of 7 items (5 potential answers each) with a total score ranging from 0 (no problem) to 4 (very serious problem). The total score is interpreted as follows: no insomnia (0–7); subthreshold insomnia (8–14); moderate insomnia (15–21); severe insomnia (22–28).

### 2.3. Statistical Analysis

Data analysis was conducted with the aid of the GraphPadPrism 8.01 statistical software package (GrapPad Company, San Diego, CA, USA). All data were analysed for normal distribution using the D’Agostino and Pearson omnibus normality test in order to determine which was the best statistical test to apply for the analysis of our data.

Since our data were not normally distributed, we chose to apply the nonparametric Chi-square test in order to check whether the frequency of the values obtained from our surveys were significantly different from the frequencies obtained with the theoretical distribution. The Chi-square test was performed to assess whether there were any differences in the scores obtained from the Insomnia Severity Index test between subjects of different sexes and between those who were healthy or diagnosed with psychiatric pathologies at the time of their medical history. The degree of correlation between insomnia in general and the presence of anxiety and depressive disorders was also assessed through the odds ratio (OR).

A descriptive data analysis was also conducted to determine the degree of insomnia in both the subjects with psychiatric illnesses and the rest of the study population.

## 3. Results

### 3.1. Insomnia Disorders in the General Population

The analysis of the data from the ISI test revealed that, within the population under evaluation, there are different subjects who suffer from insomnia. Specifically, the results obtained through descriptive data analysis revealed that 292 (70.70%; ISI 3.171, CI 2.918–3.425) have no insomnia problems, 67 (16.22%; ISI 10.90, CI 10.47–11.32) have subthreshold insomnia, 43 (10.41%; ISI 17.37, CI 16.78–17.96) have moderate insomnia and 11 (2.66%; ISI 23.00, CI 22.15–23.85) were found to have severe insomnia problems.

Furthermore, based on the results obtained on the entire population, we asked what the differences were within the male and female sample regarding insomnia-related problems (Table 1A,B).

In order to understand whether there were differences in the severity of insomnia between male and female subjects, we conducted the Chi-square test, taking into consideration the degree of insomnia obtained from the analysis of the data obtained from the ISI. The analysis of the data did not reveal any significant difference in the number of subjects with subthreshold sleep (χ^2^ = 0.2829 z = 0.5319 *p* = 0.5948), moderate (χ^2^ = 2.045, z = 1.430 *p* = 0.1527) or severe insomnia (χ^2^ = 0.4612, z = 0.6791 *p* = 0.4971), between male and female subjects.

### 3.2. Association between Sleep Disorders and Psychiatric Disorders

Data from the analysis of the ISI test regarding the number of patients with or without insomnia were also evaluated on the basis of the number of patients who had a previous or current diagnosis of anxiety or an anxious/depressive state at the time of the anamnestic assessment. Specifically, compared to persons without a history of insomnia, those with different degrees of insomnia had a higher prevalence of anxiety and depressive disorder. In detail, the Chi-square test revealed significant differences in the percentage of subjects with insomnia and a previous diagnosis of anxiety (χ^2^ = 44.84 z = 6.696 *p* < 0.0001) and anxious/depressive disorder (χ^2^ = 43.94 z = 6.629 *p* < 0.0001), compared to those with a diagnosis of anxiety and anxious/depressive disorder but no insomnia problem (Figure 1). Information on the lifetime occurrence of depression and anxiety according to sleep disorders is presented in Table 2. In addition, on the basis of the data obtained, we wanted to calculate the probability of the association of insomnia and its various degrees with the diagnosis of the psychiatric disorders that were the subject of our study (Table 3).

## 4. Discussion

On a psychosocial level, shift work is associated with reduced flexibility, less control over working conditions, a precarious work/life balance and limited time to recover energy spent at work. From a behavioural point of view, unhealthy habits can be found in these workers, such as smoking, the use of alcoholic beverages and a diet high in cholesterol, which often leads to weight gain [24].

The most important factor linking shift work to various health problems is altered sleep quality, as circadian rhythms and melatonin secretion levels are altered. Frequent consequences are therefore chronic sleep deprivation, the presence of symptoms of insomnia and excessive daytime sleepiness. Our data are consistent with what has already been established in the bibliography, about how one in four people, among shift workers, and one in three, among night workers, suffer from relevant clinical symptoms referable to SWSD sleep disorders [25].

The results showed that in our sample there is a good percentage of female (32.46%) and male (26.57%) subjects with insomnia problems. In particular, analysis of the Insomnia Severity Index returned an evident prevalence of subjects with subthreshold and moderate insomnia compared to those with severe insomnia among both female and male subjects. Furthermore, subjects suffering from insomnia disorders showed a greater likelihood of developing anxiety or anxiety–depressive disorder.

These data are in agreement with various studies that show that the presence of sleep disorders is capable of compromising normal daily activities and promoting the onset of fatigue, daytime sleepiness, deficits in cognitive performance and mood disorders [26]. In addition, studies have shown that insufficient sleep can promote the risk of developing psychiatric disorders, including anxiety and depression, or exacerbate their severity [27,28].

Total sleep loss amplifies activity within the ‘fear network’, which includes the limbic system, influencing the evaluation of emotional experiences that may result in promoting the onset of anxious behaviour [29,30]. Sleep plays such a key role in the regulation of emotions that a total loss of sleep can favour an increased activation of the amygdala, its reduced connectivity with the mPFC and an increased connectivity with the brainstem and the locus coeruleus (LC), an area strongly implicated in the regulation of sleep and wakefulness [31,32]. Insomnia, due to its ability to reduce sleep hours, would promote a reduction in emotional control by the prefrontal cortex, negatively affecting the individual subject’s responses when faced with an emotional challenge [31]. This would lead to an imbalance in the ability to manage emotional stimuli that, being negatively perceived as ‘fear’, would mark both the onset and maintenance of anxiety disorders [33,34,35].

Several substances that balance the sleep/wake state are also implicated in the emotional regulation underlying the genesis of anxiety and depression, including γ-aminobutyric acid (GABA) and serotonin (5HT). Specifically, a deficiency of GABAergic activity in the occipital cortex of patients with sleep disorders has been reported to be consistent with the hyperexcitation pattern in insomnia [36]. Similarly, serotonin, a neurotransmitter important in the regulation of attentional and cognitive processes and mood regulation, is closely related to the systems underlying the regulation of the physiological sleep–wake rhythm [37]. Alterations in these two neurotransmitters may partly explain what was highlighted in our study.

GABA is known to modulate the firing of the locus coeruleus (LC), a nucleus composed mainly of noradrenergic neurons located in the brainstem bridge, implicated as a component of the neuronal network and able to modulate the sleep–wake state [38,39]. Increased norepinephrine release by the LC is associated with a state of increased vigilance, reduced adaptive response to stress, fear and anxiety [40]. Consequently, inhibition of LC neurons could induce strong anxiolytic, sedative and hypnotic effects by improving sleep quality and sleep architecture [40,41,42]. The possible reduction in GABAergic activity in the subjects with insomnia analysed in our study, causing a reduction in LC inhibition, would therefore influence the normal adaptive response to a particularly stressful condition, explaining the finding of a correlation between insomnia and anxiety.

Another neurotransmitter important in the regulation of different physiological functions, including sleep and mood, is 5HT [43,44,45]. It has been shown that 5HT is able to modulate sleep through stimulation of the ventrolateral preoptic area of the hypothalamus [46], a region of the brain that is crucial for the induction of slow-wave sleep. Specifically, 5HT, by decreasing the frequency of excitatory events and increasing inhibitory ones projecting into the ventrolateral preoptic area, would promote both the waking and sleeping states [47]. Furthermore, serotonergic transmission is directly connected with and regulates the circadian system.

Numerous studies have shown how dysregulation of the circadian system can favour the genesis of different depressive symptoms [48,49,50]. On the other hand, the amount of endogenously generated 5HT varies throughout the day, being more concentrated during the hours of darkness than during the hours of light [51,52,53]. A reduction in 5HT levels is also one of the most widely accepted hypotheses regarding the onset of depression, so much so as to justify the use of antidepressant drugs, molecules capable of increasing serotonergic transmission, as first-line molecules for treating depressed patients.

The above could further corroborate the data obtained from our research in that the reduction of sleep hours could lead to an under-regulation of 5HT levels, which, in turn, would favour the onset of mood disorders such as anxiety–depressive disorder.

### Limitations of the Study

Although this pilot study provides an overview of how insomnia correlates with the genesis of anxiety–depressive disorders, it has several limitations. Even if the number of subjects enrolled in our study is sufficient to diagnose the presence or absence of insomnia, the number of participants diagnosed with psychiatric disorders is not sufficient to draw definitive conclusions; in this regard, we are collecting additional data to enlarge the sample. In addition, it would be desirable to carry out direct assessments of the diagnosis of anxiety and anxious depressive disorders in patients with previous insomnia problems in order to be able to investigate the influence of reduced sleep hours on the onset of psychiatric disorders.

## 5. Conclusions

In conclusion, in this pilot study, we have highlighted how anxious and depressive anxiety disorders are more present in people who suffer from altered sleep–wake rhythms, such as in those who carry out their work activities during the night hours. In particular, the presence of psychiatric disorders is not only directly related to insomnia, but also proportional to its degree of severity.

It is not infrequent, in fact, that we may be faced with particularly heavy and demanding conditions that disturb the normal homeostatic balance and the ability to cyclically adapt one’s circadian rhythms to those of work. This imbalance could therefore promote a desynchronisation of the sleep/wake cycle, increasing susceptibility to various disorders of the psycho-emotional sphere.

The further development of our pilot study and the future understanding of the exact mechanisms by which reduced sleep hours of workers periodically working night shifts may predispose them to the onset of psychiatric disorders, such as anxiety and depression, could provide valuable prevention and support tools.

Our work therefore provides a basis on how different therapeutic interventions can be undertaken in the future.

Numerous studies have in this respect highlighted the effectiveness of cognitive restructuring and behavioural activation techniques in the treatment of both depression and insomnia [54,55].

Further studies on its efficacy could make this psychological support a useful tool for those who, subjected to shift work, manifest precursor symptoms of insomnia and depression, also providing new basic results for understanding the genesis of the onset of other pathologies, apparently less important, but which may act in a concausal role on the efficiency of work performance.

Furthermore, it would be essential to provide recommendations about habits and practices that can improve the quality and quantity of sleep of the workers. These sleep hygiene recommendations can help workers improve the quality and quantity of their sleep, which can positively impact their daily functioning, mood and overall well-being.

## Figures and Tables

**Figure 1 ijerph-20-03756-f001:**
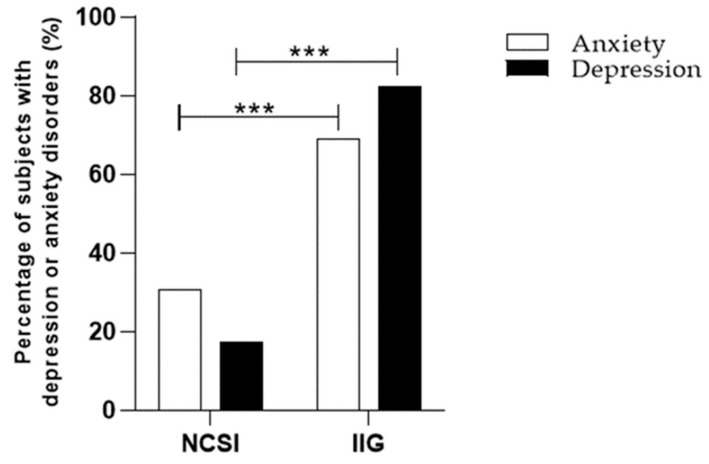
Differences in the percentage of subjects with and without insomnia in general and previous diagnosis of anxiety or depression. Results are expressed in percentages. NCSI, no clinically significant insomnia; IIG, insomnia in general; *** *p* < 0.0001 vs. NCSI.

**Table 1 ijerph-20-03756-t001:** Data on the presence or absence and degree of insomnia in male (A) and female (B) subjects.

A				
	***n***°	**%**	**Insomnia Severity Index Mean**	**95% CI**
**EVIL**	222	100	5.955	5.212–6.698
No clinically significant insomnia	163	73.42	3.08	2.747–3.413
Subthreshold insomnia	35	15.77	10.74	10.13–11.35
Moderate insomnia	19	8.56	17.37	16.43–18.31
Severe insomnia	5	2.25	22.80	21.76–23.84
**B**				
	***n***°	**%**	**Insomnia Severity Index Mean**	**95% CI**
**FEMALE**	191	100	6.984	6.099–7.869
No clinically significant insomnia	129	67.54	3.287	2.894–3.680
Subthreshold insomnia	32	16.75	11.06	10.43–11.70
Moderate insomnia	24	12.57	17.38	16.56–18.19
Severe insomnia	6	3.14	23.17	21.49–24.85

**Table 2 ijerph-20-03756-t002:** Numbers of subjects with different degrees of insomnia and anxiety or depressive disorders. Insomnia in general includes subthreshold insomnia, moderate insomnia and severe insomnia.

	No Psychiatric Disorder	Anxiety	Depression
Total Sample Size (*n* = 413)	*n*° (%)	*n*° (%)	*n*° (%)
No clinically significant Insomnia (*n* = 292)	275 (94.18)	13 (4.45)	4 (1.37)
Insomnia in general (*n* = 121)	73 (60.33)	29 (23.97)	19 (15.70)
Subthreshold insomnia (*n* = 67)	49 (73.13)	11 (16,42)	7 (10.44)
Moderate insomnia (*n* = 43)	20 (46.51)	14 (32.56)	9 (20.93)
Severe insomnia (*n* = 11)	4 (36.36)	4 (36.36)	3 (27.27)

**Table 3 ijerph-20-03756-t003:** Odds ratios for the different psychiatric disorders analysed associated with the various degrees of insomnia. The data were calculated.

	Anxiety		Depression	
	Odds Ratio	95% CI	Odds Ratio	95% CI
Insomnia in general	8.404	4.11–16.73	17.89	5.966–49.42
Subthreshold insomnia	4.749	2.086–11.43	9.821	2.946–30.58
Moderate insomnia	14.81	5.905–34.08	30.94	8.287–94.46
Severe insomnia	21.15	5.485–76.92	51.56	9.687–304.9

## Data Availability

The data is not publicly available due to privacy restrictions.

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
