# Peer review of "Evaluation of Correlation between Sleep and Psychiatric Disorders in a Population of Night Shift Workers: A Pilot Study"

_ijerph, 2023, doi:10.3390/ijerph20043756_

Round 1

Reviewer 1 Report (Previous Reviewer 1)

Authors have addressed most of my comments in the previous review report and only need to address the 4th comment as I mentioned that sentence are redundant and need to be trimmed such as ‘When we went to assess whether, in our study population, there was a correlation between the diagnosis of anxiety and/or anxious depressive disorder and the presence of insomnia, our data showed that there is a greater likelihood of developing psychiatric disorders in subjects with insomnia in its various degrees and that, their genesis is directly related to the severity of insomnia itself.’  Can be accepted for publication after the minor revision.

Author Response

Dear Reviewer,

We have prepared one table (1- peer reviewer corrections required) in which we have included the following information:

  1. the first column contains the number of comments of peer reviewer, following the sequence present in the revised paper;
  2. the second column contains the Specific Reviewer Comments and Suggestions:
  3. the third column contains the modifications made, their explanation and, above all, the number of page and lines in which these modifications have been included.

Reviewer 2 Report (Previous Reviewer 2)

The paper currently under review is a resubmitted manuscript. In your new article, the authors took into account my comments to the previous one regarding the need to extend the methodology and remove errors in Table 2. The purpose of the research was also slightly expanded. The authors write that they are conducting research in order to evaluate possible effective interventions in preventing these disorders (sleep disorders and psychiatric disorders, specifically anxiety and depression). In their conclusions, the authors suggest that such intervention should consist in psychological support, cognitive restructuring and behavioural activation techniques as a useful tools for those who, subjected to shift work, manifest precursor symptoms of insomnia and depression. Such actions may be considered justified, however, when the trigger of sleep disturbances (circadian rhythm disorders due to night/shift work) is known, interventions should take into account specific sleep hygiene recommendations for workers and suggestions concerning such changes to the organization of shift work that would reduce the possibility of circadian rhythm disturbances.

Author Response

Dear Reviewer,

We have prepared one table (1- peer reviewer corrections required) in which we have included the following information:

  1. the first column contains the number of comments of peer reviewer, following the sequence present in the revised paper;
  2. the second column contains the Specific Reviewer Comments and Suggestions:
  3. the third column contains the modifications made, their explanation and, above all, the number of page and lines in which these modifications have been included.

This manuscript is a resubmission of an earlier submission. The following is a list of the peer review reports and author responses from that submission.

Round 1

Reviewer 1 Report

The manuscript ‘Association between sleep and psychiatric disorders in a population of night shift workers’ aims to investigate the anxious and depressive anxiety disorders in people who suffer from sleep disorders specially shift workers. They used Insomnia Severity Index questionnaire and statistically analyzed the difference between group with or without sleep disorders in different subgroup of individuals. Authors pointed out that anxious and depressive anxiety disorders are more pronounced in people who suffer from altered sleep-wake rhythms. I think the design and aim are okay but the manuscript needs tremendous improvement before it can be published on IJERPH. My major comments are attached as below:

1.       Since the authors aim to investigate the association between sleep disturbance and anxiety and depression in the group pf shift workers, authors should first investigate how and sleep disturbance contributes to depression/anxiety. Shift workers are a group of individuals that has been studies that have negative effects on their mental health and metabolic condition, the design of the study has been compromised by the skipping the correlation between sleep disturbance and psychiatric disorders, should be at least introduced and discussed in details.

2.       As I mention in the 1st comment, the relationship between sleep disturbance and depression/anxiety has been widely studies and if authors want to specify the specificity in shift workers, they should at least involve more individuals to exclusively study the association of sleep disturbance and why it increases the incidents of psychiatric disorders in general populations.

3.       Authors aims to figure out the anxiety/depression and degree of insomnia in shift workers, they should pre-introduce the criteria for anxiety/depression diagnosis in the individuals, whether or not the patients taking medications or not, indicating some important information missing in the manuscript for the study, needs to be revised carefully for these details.

4.       I found some sentence are redundant and difficult to understand in the introduction and discussion. Such as ‘when we went to assess whether, in our study population, there was a correlation between the diagnosis of anxiety and/or anxious depressive disorder and the presence of insomnia, our data showed that there is a greater likelihood of developing psychiatric disorders in subjects with insomnia in its various degrees and that, their genesis is directly related to the severity of insomnia itself.’  Conclusive sentence like this should be trimmed for better understanding.

Thanks.

Reviewer 2 Report

The aim of the presented study is to evaluate the association between sleep disorders and psychiatric disorders, specifically anxiety and depression, on a group of workers of both sexes who performed different work activities involving night shifts for at least 50 nights per year. This issue is important because stress, depression, and anxiety are the second most common work-related health problem. That is why actions aimed at understanding the mechanisms of their formation are so important. The research was properly planned. However, I have a few remarks regarding the method of presenting the obtained results and discussing them.

In the abstract is the following conclusion: “we have highlighted how anxious and depressive anxiety disorders are more pronounced in people who suffer from altered sleep-wake rhythms”. This is the result of the study, not a conclusion.

Despite the fact that standardized tools are used in the diagnosis of anxiety and depression, in the study conducted by the authors, the presence of a clinical diagnosis of current or past anxiety and/or anxiety-depressive state was carried out solely on the basis of an anamnesis. It is not known whether the assessment of the mental state was carried out by the same person, and if not, whether structured interviews and uniform criteria for assessing the severity of anxiety symptoms and depressive symptoms were used.

Despite the fact that epidemiological and environmental studies indicate that anxiety and depressive disorders are generally more common among women than among men, this study did not attempt to examine whether such differences exist. Only whether there were differences in the severity of insomnia between men and women was analyzed.

In Table 2 error has crept in:                                                                                                                                   No psychiatric     Anxiety    Depression

                                                              disorders

                                                           n (%)              n (%)              n (%)

Subthreshold insomnia (n=67)      49 (73.13%)    11 (16,42%)   7 (59.11%)

                                                                                                   It should be                                                                                                             10,44%

The authors focused on the analysis of the relationship between anxiety or depression and the occurrence of insomnia, while in the literature one can find data indicating a multifactorial etiology of anxiety or depression in the working population. The role of psychosocial work characteristics is significant, as indicated by the relationship of some work characteristics, such as high psychological job demands or conflict with supervisors with depression, and low decision latitude and low social support with depression.

Reviewer 3 Report

This paper details the results of a study on subjective insomnia reports and medical-history anxiety/depression reports among 222 male and 191 female shift workers. Unsurprisingly, the results indicated that about 30% of the participants had insomnia and that there was an association between disturbed sleep and history of depression and/or anxiety. Thus, I am uncertain that this paper contributes meaningfully to the existing literature, especially since there are relatively recent review papers that cover these topics more comprehensively (see references 1 and 2 below).

However, beyond the questionable scientific value of the present submission, there are other issues that argue against publication:

1.       The title of the submission suggests that the focus will be on shift workers, and while this is in fact somewhat true, the Introduction section doesn’t seem to place much emphasis on the copious amount of existing literature on shift work and sleep. In particular, the authors fail to even mention Shift work sleep disorder, despite the well-known fact that “a substantial percentage of shift workers develop shift work disorder, a circadian rhythm sleep disorder characterized by excessive sleepiness, insomnia, or both as a result of shift work” (see reference 3 below). This is a serious omission.

2.       Furthermore, the Introduction section in general should more concisely summarize the literature that is most relevant to the topic at hand.  For instance, the section on stress (lines 63-76) is fairly tangential in the present context and probably should be deleted altogether.

3.       The methodology is also less than optimal in terms of the manner in which depression and anxiety status were determined.  Since subjects are already completing a questionnaire on insomnia (via the Insomnia Severity Index), why not also have them complete questionnaires on depression and anxiety (via the Patient Health Questionnaire and General Anxiety Disorder inventory--PHQ-9 and GAD-7) rather than relying upon medical-history reviews? If the authors had elected to do so, they would have had more accurate data on self-rated insomnia, anxiety, and depression, all during the most recent 2-week period, and the additional questionnaires would have added only 5 minutes to the assessment session. Instead, they collected recent sleep information directly from the volunteers while relying on historical records and physician assessment of the other conditions, and this is a questionable strategy.

4.       Given that the focus of the paper was on all 3 conditions (insomnia, anxiety, and depression), why did the authors assess gender differences only on the insomnia dimension? Testing for gender differences in anxiety and depression as well would have been an important component of the present research since it would have been helpful to know whether the absence of anticipated gender differences on sleep were corroborated by an unexpected absence of gender differences in anxiety/depression.  In general, females typically express more difficulty in all 3 areas.

5.       The Discussion section contains far too much tangential information that appears “out of place” within the present context.  Most notably, the details on neurological/physiological factors (lines 215-252) should be either significantly streamlined or deleted entirely since the authors did not collect physiological measures of any sort while the circadian/shift-work issues should have been more clearly emphasized.

6.       Finally, the statement on lines 207 and 208 implies that the sleep problems identified in the present study CAUSED the mental health problems, and this conclusion is unsupportable in the present context.  In fact, there is an ongoing debate within the published literature regarding bidirectional relationship between mental disorders and sleep.  Does sleep cause anxiety and depression, or do anxiety and depression cause sleep disorders?  The answer to this question remains controversial, and the authors have not resolved it especially within the present context.

Given the issues noted above in combination with the overall questionable scientific value of the findings of the present research, I cannot recommend publication.

References

1.       Brito RS, Dias C, Afonso Filho A, Salles C. Prevalence of insomnia in shift workers: a systematic review. Sleep Sci. 2021 Jan-Mar;14(1):47-54. doi: 10.5935/1984-0063.20190150. PMID: 34104337; PMCID: PMC8157778.

2.       Cox RC, Olatunji BO. A systematic review of sleep disturbance in anxiety and related disorders. J Anxiety Disord. 2016 Jan;37:104-29. doi: 10.1016/j.janxdis.2015.12.001. Epub 2015 Dec 21. PMID: 26745517.

3.       Wickwire EM, Geiger-Brown J, Scharf SM, Drake CL. Shift Work and Shift Work Sleep Disorder: Clinical and Organizational Perspectives. Chest. 2017 May;151(5):1156-1172. doi: 10.1016/j.chest.2016.12.007. Epub 2016 Dec 21. PMID: 28012806; PMCID: PMC6859247